# Structural and biochemical insights into the mechanism of the Gabija bacterial immunity system

Yanwu Huo [1,4] ✉, Lingfei Kong[1,2,4], Ye Zhang[1,2,4], Min Xiao[1], Kang Du[1], Sunyuntao Xu[1], Xiaoxue Yan [1], Jun Ma [3] ✉ & Taotao Wei [1,2] ✉

The Gabija system is a newly discovered bacterial immune system that consists of GajA and GajB. Here we report the cryo-EM structure of the Gabija complex from *Bacillus cereus VD045* at 3.6 Å, which provides the direct evidence of interactions between GajA and GajB. The Gabija complex is an octameric ring structure with four GajA and four GajB. GajA is an OLD nucleases family protein, while GajB belongs to the SF1 helicases. The Gabija complex has sequence-specific DNA nuclease activity and prefers circular rather than linear DNA as substrate, its activity is more sensitive to concentrations change of nucleotides compared to GajA alone. Our data suggest a mechanism of Gabija immunity: the nuclease activity of Gabija complex is inhibited under physiological conditions, while it is activated by depletion of NTP and dNTP upon the replication and transcription of invading phages and cleave the circular DNA to prevent phage DNA replication.

Bacteria have developed sophisticated defense systems to protect themselves from the invasion of bacteriophages. Different defense strategies have been employed to prevent the proliferation of phages. Examples include preventing phage DNA from entering cells, damaging the invading DNA and promoting host cell death. Numerous bacterial defense systems employ the strategy of degrading the invading DNA. This is achieved through sequence-specific DNA cleavage, such as CRISPR/Cas and R-M systems[1,2]. By understanding these mechanisms of immune defense, powerful molecular biology tools or gene editing tools have been developed, including the well-known restriction endonuclease, CRISPR/Cas9.

An increasing number of defense systems have been discovered recently[3–8]. Doron et al. predicted computationally and validated experimentally 10 previously unknown bacterial immune defense systems[3]. Among these newly discovered systems, the Gabija system is the most widespread, accounting for 8.5% of the sequenced bacterial and archaeal genomes (4360 genomes). The Gabija system consists of two genes, GajA and GajB. A previous study showed that the Gabija system from *Bacillus cereus VD045* exhibits strong immunity against bacteriophages phi29, rho14, phi105 and SpBeta, indicating that the Gabija system is sufficient for bacterial immunity against certain bacteriophages[3]. GajA is a sequence-specific DNA-cleaving enzyme in the presence of $Mg^{2+}$, its activity is completely inhibited by the high concentration of NTP and dNTP[9]. Prior research indicates that GajA plays an essential role in Gabija immunity system by sequence-specific DNA cleveage regulated by nucleotide concentration. However, the immunity of *E. coli* to bacteriophage T7 requires the whole Gabija gene cassette, indicating that the other gene of the Gabija system, GajB, is indispensable.

In this work, we purify and characterize the Gabija complex (GajA and GajB) from *B. cereus VD045* and present the structure of the Gabija complex at 3.6 Å resolution. The resolved structure shows that the Gabija complex adopts an octameric ring structure with four GajA and four GajB. This observation provides direct evidence of the interaction between GajA and GajB in the Gabija defense system. GajA is an OLD (overcoming lysogenization defect) nucleases family protein

[1]National Laboratory of Biomacromolecules, Institute of Biophysics, Chinese Academy of Sciences, 15 Datun Road, Chaoyang District, Beijing 100101, China. [2]School of Biological Sciences, University of Chinese Academy of Sciences, 19 Yuquan Road, Shijingshan District, Beijing 100049, China. [3]Institute of Infectious Diseases, Shenzhen Bay Laboratory, Gaoke Innovation Center, Guangqiao Road, Guangming District, Shenzhen, Guangdong 518132, China. [4]These authors contributed equally: Yanwu Huo, Lingfei Kong, Ye Zhang ✉e-mail: huoyw@ibp.ac.cn; majun@szbl.ac.cn; weitt@ibp.ac.cn

containing an ATPase domain and a Toprim (topoisomerase/primase) domain. The ATPase domain of GajA encodes a conserved active site and adopts a productive conformation with the ATP-binding cassette (ABC) signature sequence orienting toward the active site of the opposing subunit in close approach to the P loop and bound nucleotide, while the Toprim domain has a conserved nuclease active site. On the other hand, GajB belongs to superfamily 1 (SF1) helicases and shares the essential features of this family conserved in spatial distribution. Our data further suggest that the Gabija complex has the sequence-specific DNA endonuclease activity and prefers supercoiled circular DNA as its substrate. The cleavage activity of Gabija complex is completely inhibited with 0.5 mM ATP towards the linear DNA substrate, while the cleavage activity on supercoiled circular DNA is still preserved at 0.5-2 mM ATP and inhibited by much higher concentrations of nucleotides. Together our data provide important insights into the structure and mechanisms of the Gabija bacterial immunity system.

## Results

### The overall structure and structural organization of the Gabija complex

The Gabija complex from *B. cereus VD045* was heterogeneously expressed in *E. coli* and purified by chromatography (Supplementary Fig. 1a). Size exclusion chromatography coupled with multi-angle light scattering (SEC-MALS) revealed that the Gabija complex forms a stable hetero-octamer in solution (Supplementary Fig. 1b). In contrast, individual GajA and GajB exist in the form of tetramer and monomer, respectively (Supplementary Fig. 1c, d). Therefore, it is inferred that the octamer form of the Gabija complex is the minimum work unit of the Gabija system. Both GajA and GajB are essential for anti-phage activity of Gabija system; deletion of either gene leads to the loss of Gabija immunity revealed by phage resistance assay in vivo (Supplementary Fig. 2). Subsequently, the structure of Gabija complex is solved by the cryo-EM single particle analysis at a resolution of 3.6 Å (Supplementary Fig. 3 and Supplementary Table 1). The whole Gabija complex resembles an octameric ring with three 2-fold symmetry axes. The three dimensions of the octameric ring are 171 Å × 143 Å × 91 Å; The inner diameter of the ring is 35 Å at the narrowest region. Each octamer containing four GajA and four GajB (Fig. 1a, b and Supplementary Fig. 4a, b).

SEC-MALS results show that individual GajA and GajB exist in the form of tetramer and monomer respectively (Supplementary Fig. 1c, d), supporting the hypothesis that the octameric ring of Gabija complex is formed though two steps: the tetramer formation of GajA and subsequent octamer formation of Gabija complex. The tetramer formation of GajA leads to the interfaces between GajA1-GajA2 and GajA1-GajA3 (Fig. 1c, d). The A1-A2 interface involves symmetric interactions, the contact sites are mainly related to ATPase domain of GajA. They are α2, β6, the loop between α5 and β8, the loops on both sides of the β6 (Fig. 1c). The A1-A3 interface involves symmetric interactions. These contacting sites spread both the ATPase and Toprim domains of GajA. They are α5, α6, α7, α8, α10, α15, α16, β13, the loop between α14 and α15, the loop between α10 and β14, the loop between α8 and β12, the loops on both sides of the β13, the loop between α7 and β11, the loop between α9 and β10, the loop between α1 and β3, the loop between α6 and β8 (Fig. 1d). The octamer formation of Gabija complex leads to the interfaces between GajA1-GajB1, GajA1-GajB2 and GajB1-GajB2 (Fig. 1e, f). The interactions within GajA1-GajB1 are mediated by the ATPase domain of GajA and the 1A, 1B domains of GajB. The α3 helix, the loops on both sides of the α3 helix, the loop between α5 and β8 and the α4 helix of GajA are involved in this contact interface. In GajB, the interface contact sites are α4, α7, the loop between α2 and β2 and the loop between α3 and β3 (Fig. 1e). The flexible CTR region (C-terminal region, residues 224-416) of GajB could not be traced in the electron density map. To determine the interactions between GajA and CTR

region of GajB, two constructs of GajB have been made, the NTR region (N-terminal region, residues 1-223) and the CTR region. Size exclusion chromatography demonstrates that the NTR region of GajB indeed binds to GajA, while the CTR region of GajB doesn't bind to GajA (Supplementary Fig. 5a, b). Interface A1-B2 is mediated by the β5, β6, the loop between β5 and β6 of GajA ATPase domain and by the loop between α3 and β3 of GajB 1A domain. Interface B1-B2 is mediated by the α5 and the loops on both sides of the α5 (Fig. 1f).

### Structural organization of GajA and GajB

GajA is an OLD nucleases family protein based on sequence homology, and contains two domains: an N-terminal ATPase domain (1-351 amino acid) and a C-terminal Toprim domain (352-578 amino acid) (Fig. 2a). The ATPase domain contains an eleven-stranded β-sheet with the order β6-β5-β4-β1-β2-β11-β10-β3-β9-β8-β7. This β-sheet folds into two layers at the β2 and β11 strands and a central α helix α1 inserts into it, forming a sandwich-like structure. Helices α2, α3, and α4 flank the sandwich, capping the N terminal of α1. Helices α5 and α6 lie at the bottom of the sandwich, making a four-layer structure.

The Toprim domain contains a four-stranded β-sheet with the order β13-β12-β14-β15. The helices α11 and α12 rest on the top of β15 within the same plane of the four-stranded β-sheet. The helices α7, α9, and α10 lie on the left side of the β-sheet plane, interacting with the β11, β10 and the loop within the β10-β9 of the ATPase domain. The helices α8, α13, α14, α15, α16, α17, and α18 lie on the right side of the β-sheet plane.

GajB subunit belongs to SF1 helicase based on sequence alignment, it contains four structural domains 1A (1-72, 157-223 AA), 1B (73-156 AA), 2A (224-305 AA, 417-494 AA) and 2B (306-416 AA) (Fig. 2b). Despite being present in the cryo-EM protein sample, domain 2A and 2B could not be traced in the electron density map, demonstrating that this region is quite flexible in the Gabija complex.

### Essential features of SF1 helicase of GajB

Sequence alignment reveals that 11 typical sequences commonly found in SF1 helicases are well conserved in GajB motifs (I-VI, Ia,Ib, Id, IVa and IVc) (Fig. 2c and Supplementary Fig. 6). The seven sequence motifs (I, Ia, II-VI) and Q motif conserved in SF1 helicase are involved in ATP binding as reported. In our structure, the ATP binding motifs (I, Ia, II, III) gather together and form the most conserved region (Fig. 2d). The Q motif is supposed to be situated in the α1 helix of the GajB structure, while the Q motif is missing in α1 helix. The adenine base is selected explicitly by Q motif, the missing of the Q motif results in the loss of selectivity of ATP. The GTPase activity of GajB is observed indeed in the following biochemical assay, but the GTPase activity of GajB appears much weaker than the ATPase activity, which indicating that the ATPase domain of GajB still has a higher affinity for adenine than guanosine. (Supplementary Fig. 7b, e). The good conservation of sequence alignment and spatial distribution of typical motifs imply that GajB may employ a similar mechanism for ATP hydrolysis as the SF1 helicases family.

### GajA Toprim domain have a conserved nuclease active site

There is a highly conserved region (E379, E383, D432, D434, D511, D513, K541) in the Toprim domain (Fig. 3c). By structural alignment with OLD CTR structure from *Burkholderia pseudomallei* (OLD_Bp), this region is determined to be the site of nuclease activity (Fig. 3a). GajA residues E379, E383, D432, D434, D511, D513, align well with the OLD_Bp metal binding residues E400, E404, D455, D457, T506 and E508. GajA residue K541 roughly aligns with the OLD_Bp K562 (Fig. 3b). K562 is the catalytic amino acid that plays a role in stabilizing the negative charge of transition state of the phosphoryl transfer reaction and protonate the leaving group. The spatial conservation of the catalytic site and good alignment with OLD_Bp may imply that GajA employs conserved

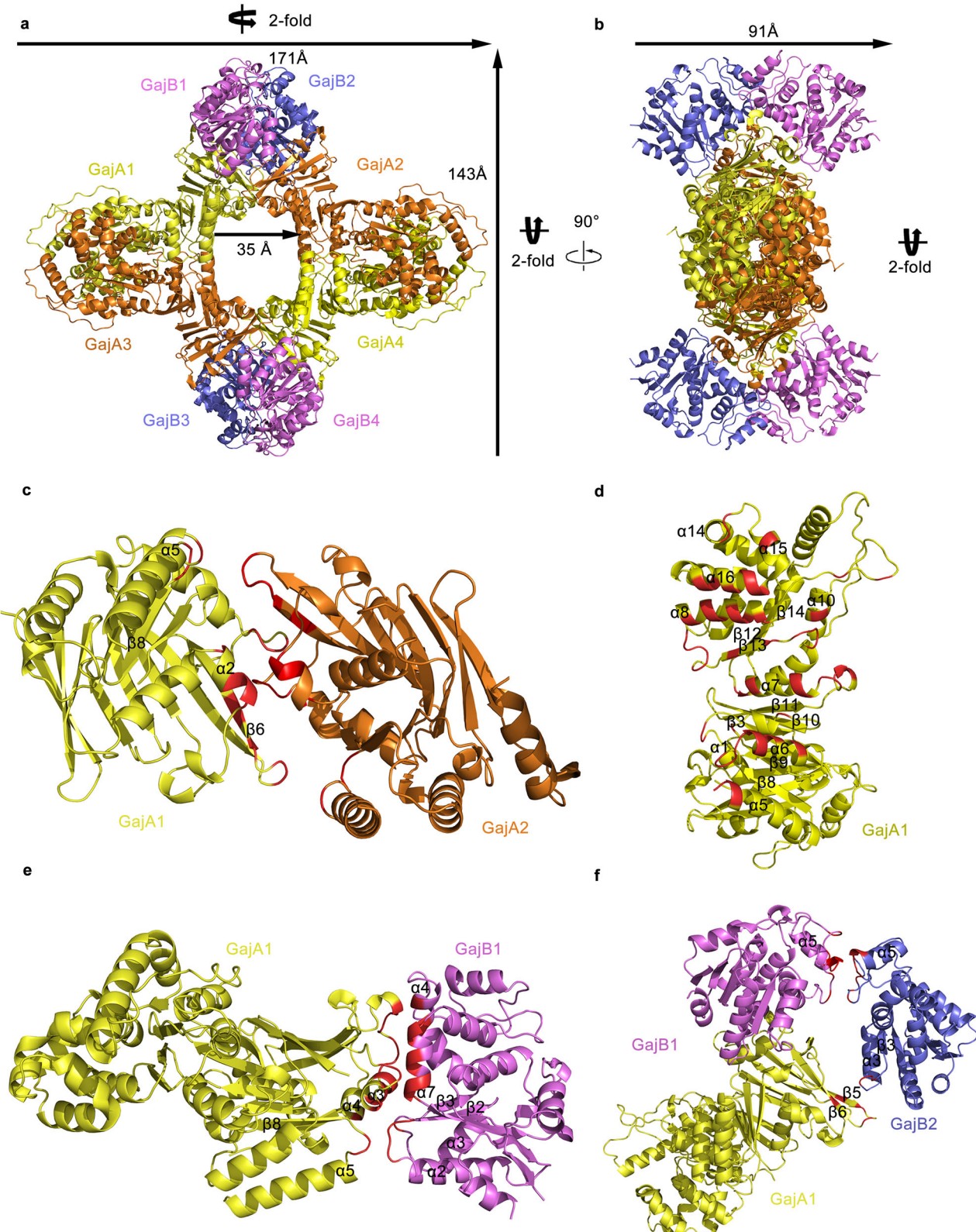

**Fig. 1 | Overall structure of the Gabija complex and interactions between the subunits of the Gabija complex. a** The top views of the Gabija complex octameric ring structure. **b** The side views of the Gabija complex octameric ring structure. The three 2-fold symmetry axes and the measurement of the octameric ring dimensions are marked. **c**, **d** The tetramer formation of GajA leads to the interactions between GajA1-GajA2 and GajA1-GajA3 (interface A1-A2 in panel **c** and interface A1-A3 in panel **d**), the contacting residues in the GajA1-GajA3 interface are symmetric related, for clarity, only subunit D is displayed. The octamer formation of Gabija complex leads to the interactions between GajA1-GajB1, GajA1-GajB2 and GajB1-GajB2; (interface A1-B1 in panel **e**, interface A1-B2 and B1-B2 in panel **f**). The four GajA are colored yellow and orange. The four GajB are colored blue and purple. The contacting residues in the interface are colored red.

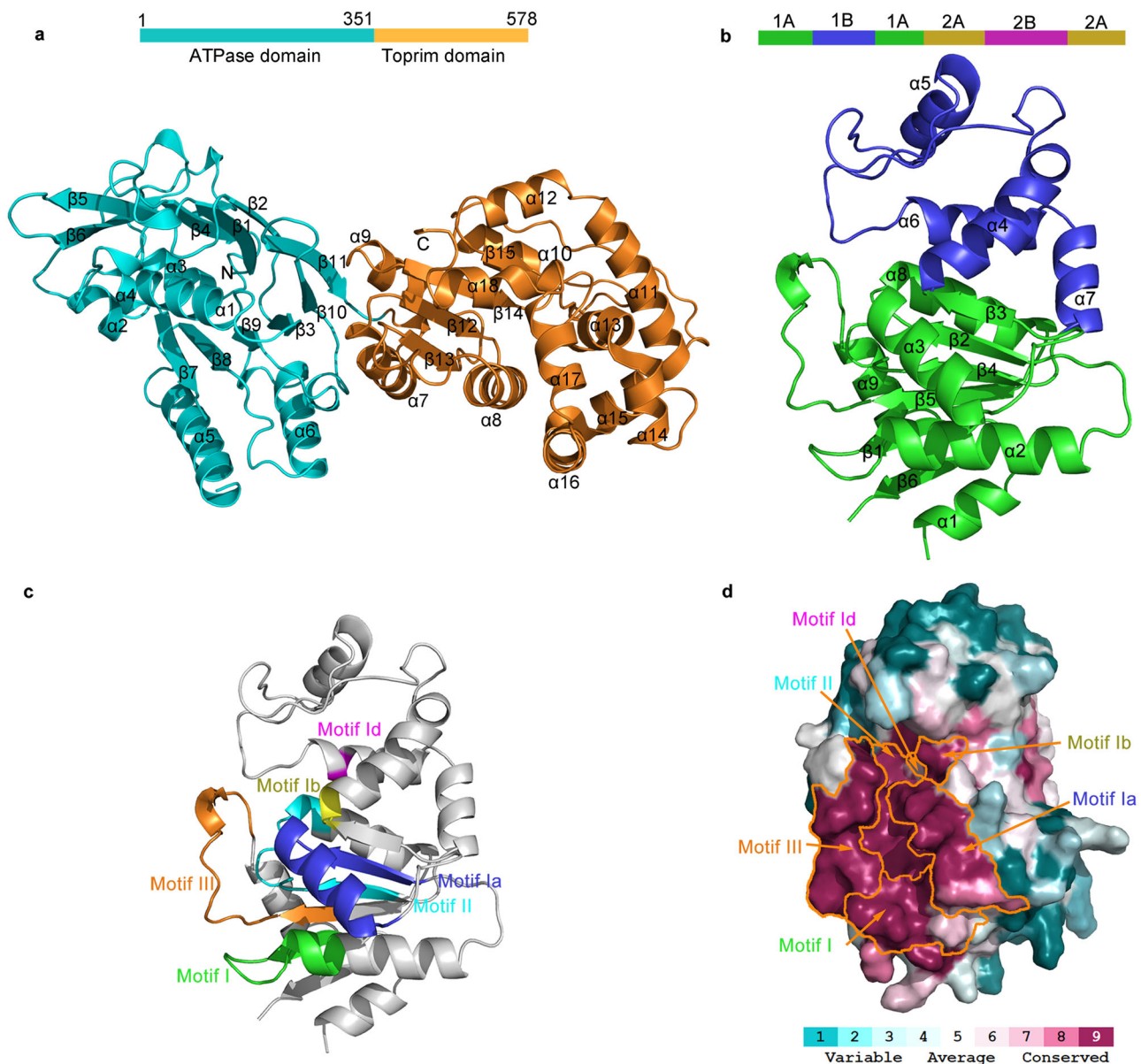

**Fig. 2 | Structural analysis of GajA and GajB. a** Structural organization of full-length GajA. GajA contains two domains: an N-terminal ATPase domain (1-351 amino acid) and a C-terminal Toprim domain (352-578 amino acid), as illustrated by a schematic drawing of the structural domain. The number of boundaries is labeled. ATPase domain is colored cyan; Toprim domain is colored orange. **b** Structural organization of GajB. GajB contains four structural domains 1A (1-72, 157-223 AA), 1B (73-156 AA), 2A (224-305 AA, 417-494 AA) and 2B (306-416 AA) as illustrated by schematic drawing of the structural domain. 1A domain is colored green, 1B domain is colored blue, and the electron density of 2A and 2B domains is missing due to structural flexibility. **c, d** Dissection of the SF1 helicase of GajB. **c** Mapping of the SF1 helicase motifs onto the GajB structure. **d** Surface conservation in the GajB structure by ConSurf server[21]. Residues are colored from magenta to cyan with descending order of conservation.

catalytic machinery and a similar two-metal mechanism for nuclease cleavage[10].

## The molecular mechanism of substrate selectivity of GajA

Based on sequence alignment, the GajA is homologous to the OLD family nucleases (OLD_Bp and OLD_Ts)[10,11]. The DNA cleavage activity of the homologous OLD nucleases is non-specific, while GajA exhibits sequence-specific DNA endonuclease activity. To explore the mechanism of substrate selectivity of GajA, we analyze the structure of GajA and carry out biochemical verification. The structure analysis reveals two unique lysine-rich structure elements in the GajA: an α-helix (residue 474-493) and a flexible loop (residue 436-442). Two unique lysine-rich structure elements are divided into three patches according to the position relative to the active center: patch 1 (K436,

K438, K439, K441, K442), patch 2 (R481, R485, K487, K491) and patch 3 (K474, K476, K477, K478, K479). Patch 1 and 3 are just around the active center. In the contrary, patch 2 is on the opposite side of the active center (Fig. 4a). To examine whether the basic residues on three patches contribute to the substrate selectivity of GajA, we made three mutants, GajA$_{mut1}$ (K436A, K438A, K439A, K441A, K442A), GajA$_{mut2}$ (R481A, R485A, K487A, K491A) and GajA$_{mut3}$ (K474A, K476A, K477A, K478A, K479A). Non-specific DNA cleavage activity is only detected in the GajA$_{mut1}$(Fig. 4b, c). These data suggest that the lysine-rich basic patch 1 of loop (residue 436-442) is responsible for the selection of sequence specific DNA digestion of GajA. Due to the lack of structural information of GajAB complex with DNA substrates, it could only be inferred from biological mutation experiments, the selection of sequence specific DNA digestion of GajA may be related to interactions

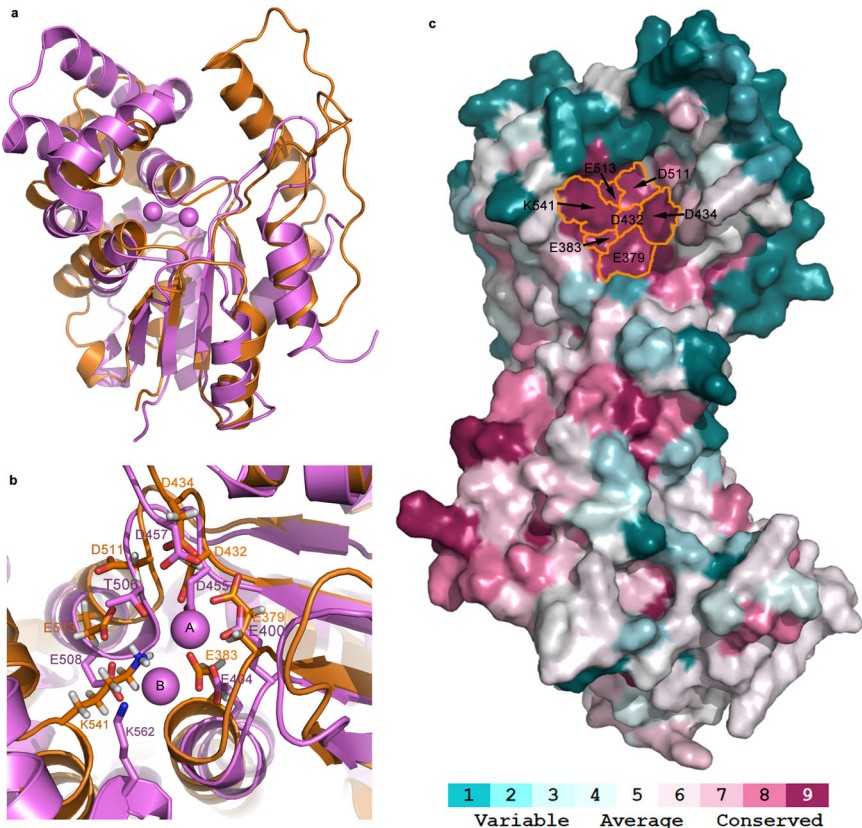

**Fig. 3 | GajA Toprim domains have a conserved nuclease active site. a** Structural alignment of GajA Toprim domains from *Bacillus cereus* VD045 (orange) and *Burkholderia pseudomallei* (OLD_Bp, PDB: 6NK8; purple) with the RMSD of 3.2 Å. **b** Close-up view of active sites of nuclease of GajA Toprim domains. Coloring as in **a**. Bound magnesium ions in 6NK8 are shown as purple spheres marked with A and B. The key residues in the active sites are labeled. **c** Surface conservation of active site residues in the GajA structure by ConSurf server[21]. Residues are colored from magenta to cyan in descending order of conservation.

between the phosphate backbone of the DNA substrate palindrome structure and GajA$_{mut1}$.

## GajA ATPase domains have a conserved active site and adopt a productive conformation

The ATPase domain of GajA is related to ABC proteins family based on the sequence alignment and structure comparison. Structure alignment by Dali[12] with OLD nuclease from *Thermus Scotoductus* (OLD_TS, PDB: 6p74, Z-score = 20.5, RMSD = 3.4 Å) reveals that they have similar overall structure and identifies six conserved sequence motifs (P loop, Q loop, the ABC signature sequence, Walker B, D loop and H loop) in GajA that contribute to ATP binding and hydrolysis (Supplementary Fig. 8a). These six sequence motifs cluster together to form the most conserved region in the ATPase domain of GajA (Supplementary Fig. 8b). The conservation of sequence and spatial distribution in the active site of ATPase imply that the ATPase domain of GajA utilizes a conserved catalytic machinery for ATP binding and hydrolysis. The dimerization of nucleotide binding domain (NBD) of ABC ATPases is necessary for ATP binding and hydrolysis, which is mediated by sandwiching two ATP molecules at the interface between the associating subunits. After dimerization of NBD, the catalytic machinery for productive ATP hydrolysis is formed, which is not only composed of the P loop, Q loop, Walker B, and H loop from one subunit, but also composed of the ABC signature sequence and D loop from the other opposing subunit (Supplementary Fig. 8c). Structure analysis of the dimer of ATPase domain of GajA showed that GajA adopt a productive ATP hydrolysis state like chromosome partition protein SMC (PDB:5xg3). In this state, the ABC signature sequence is oriented toward the active site of the opposing subunit in close approach to the P loop and bound nucleotide (Supplementary Fig. 8c).

This orientation of the ATPase domains within GajA dimer is significant different from the OLD nuclease (PDB:6p74), which is the only full-length structure of OLD nuclease resolved so far. Superposition of ATPase domains of GajA dimer with OLD nuclease shows that one subunit from each ATPase domain dimer aligns well (Supplementary Fig. 8e). However, the second GajA subunit is rotated 180 degrees relative to OLD nuclease around the dimer axis (Supplementary Fig. 8f). The D loop in the second GajA subunit is far away nearly 48 Å from the its counterpart in OLD nuclease (Supplementary Fig. 8f). This suggests that the conformations of the ATPase domains within the GajA dimer and OLD nuclease dimer are two different function state, these two conformations would inter-change during the ATP binding and hydrolysis cycle.

## Gabija complex prefers to cleave supercoiled circular DNA substrate to linear DNA

In concert with a previous study showing that GajA is a sequence-specific and metal-dependent DNA nuclease[9], the GajA protein we purified showed the same cleavage activity on 955 bp linear DNA substrate and supercoiled plasmids containing recognition sequence (5′-AATAACCCGGATATT-3′). Intriguingly, the Gabija complex we purified showed substrate preference on supercoiled circular DNA rather than linear DNA substrate (Fig. 5a, b). The cleavage activity of the Gabija complex on linear DNA substrate decreased drastically as only about 30% of the linear DNA substrate is digested by Gabija complex (Fig. 5c). The cleavage activity of the Gabija complex on supercoiled plasmid is not affected at all compared to GajA protein alone, almost all the supercoiled plasmid was completely digested by Gabija complex (Fig. 5d). Together, these data suggest that the Gabija complex has

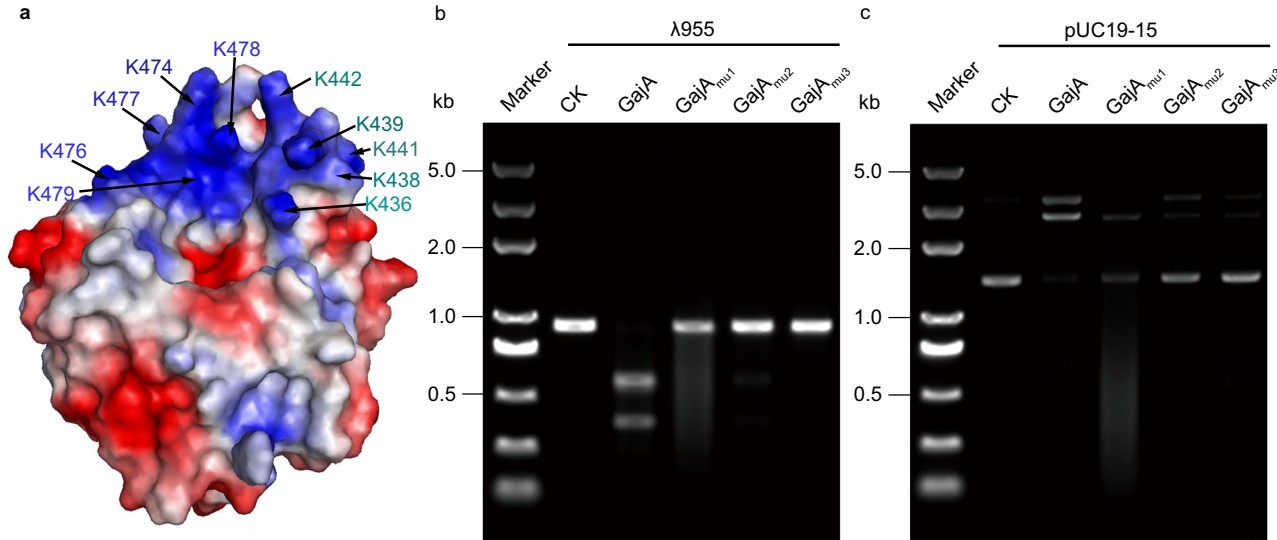

**Fig. 4 | The molecular mechanism of substrate selectivity of GajA. a** Electrostatic potential of GajA Toprim domains. The residues in patch 1 (K436, K438, K439, K441, K442) and patch 3 (K474, K476, K477, K478, K479) just around the active center is labeled. **b** Nuclease activity of wild-type GajA and mutant protein on λ955 linear DNA substrate; **c** Nuclease activity of wild-type GajA and mutant protein on supercoiled plasmids substrate; CK, control reaction without GajA. λ955 is a 955 bp fragment from lambda DNA genome with the recognition sequence (5′-AATAACCCGGATATT-3′). The pUC19-15 plasmid contains the cleavage recognition sequence (5′-AATAACCCGGATATT-3′). GajA$_{mut1}$ (K436A, K438A, K439A, K441A, K442A), GajA$_{mut2}$ (R481A, R485A, K487A, K491A) and GajA$_{mut3}$ (K474A, K476A, K477A, K478A, K479A). All the assays are repeated at least 3 times with similar results. Source data are provided as a Source Data file.

stronger substrate selectivity with the presence of GajB compared to GajA alone.

## The cleavage activity of Gabija complex is regulated by nucleotides

The structural analysis demonstrate that Gabija complex has two ATPase active site, one is in the ATPase domain of GajA, and the other one is in the interface of A1 and A2 domain of GajB. We measured the ATPase activity of purified GajA, GajB, and the Gabija complex by monitoring the amount of free phosphate released. We observed that the Vmax of GajA, GajB and Gabija complex activity of ATP hydrolysis is 4.088 mM/min, 0.953 mM/min and 7.801 mM/min, respectively (Supplementary Fig. 7a–c). When GajA and GajB form a complex, the positive synergistic effect of ATP hydrolysis capacity of the complex is observed, which suggests that GajB may be the regulator of GajA, and the interaction between them may affect their ATPase activity. We also observed that the Vmax of GajA, GajB and Gabija complex activity of GTP hydrolysis is 1.364 mM/min, 0.218 mM/min and 1.829 mM/min, respectively (Supplementary Fig. 7d–f). In the case of GTP substrate, there is also a positive synergistic effect, although it is weaker than ATP subtrate, which is consistent with the affinity of the substrate.

To further investigate the synergy of two the ATPase active sites in regulating Gabija nuclease activity, we conducted alanine screening of conserved residues in two ATPase active sites of the Gabija complex. Two mutants are constructed, one is H320A of GajA, and the other one is D162A and E163A of GajB. Both mutations in two ATPase active sites of Gabija complex abolish the anti-phage activity of the Gabija system (Supplementary Fig. 2). The most striking phenotype observed is that the AB$_{mut}$ Gabija complex is much more active in degrading linear DNA than the wild-type protein (Supplementary Fig. 7g). This result further revealed that the ATPase activity of GajB is related to the substrate selectivity of the Gabija complex, making Gabija complex prefer supercoiled circular over linear DNA. The loss of circular DNA substrate preference in AB$_{mut}$ Gabija complex may result in abolishing the phage resistance of Gabija system in vivo (Supplementary Fig. 2). D162A and E163A mutations in GajB partially relieve the inhibition of ATP and GTP on Gabija complex endonuclease activity only in the presence of optimum supercoiled plasmid substrates (Supplementary

Fig. 7g, h), H320A mutation in GajA show no effect on relief of inhibition of ATP and GTP on Gabija complex endonuclease activity both in linear and supercoiled DNA substrate, suggesting that the ATPase domain in the GajB may regulate the nuclease activity by sensing nucleotide change.

Next, we sought to test the effect of ATP or GTP on the nuclease activity of the Gabija complex. In the presence of linear DNA substrate, the low concentration (0.05 mM) of ATP/GTP both promote the nuclease activity of the Gabija complex; The high concentration (0.1 mM and above) of ATP/GTP both inhibit the nuclease activity of Gabija complex (Fig. 6a, b). In the presence of supercoiled plasmids substrate, the nuclease activity of the Gabija complex on the plasmid is significantly inhibited with the increased concentration of ATP and GTP. When the concentration of ATP reached 2 mM, the nuclease activity of the Gabija complex was entirely suppressed (Fig. 6c, d). We further measure the effect of different nucleotides on the cleavage activity of the Gabija complex. The results reveal that the cleavage activity of the Gabija complex is significantly inhibited by different NTP, dNTP, ADP, but not AMP. An interesting observation is that dNTP is more effective than NTP in the presence of pUC19-15 substrate, this implies that the depletion of dNTP caused by DNA replication may play a more important role in regulating cleavage activity of the Gabija complex (Fig. 6e). Apparently, the nuclease activity of the Gabija complex is negatively regulated by nucleotides with the exception that the low concentration of ATP/GTP promote the linear DNA substrate nuclease activity of the Gabija complex. We also measure the effect of non-hydrolysable analog of ATP on the cleavage activity of the Gabija complex (Supplementary Fig. 9). The results reveal that the cleavage activity of the Gabija complex is also inhibited by AMP-PNP, this indicates the binding of ATP is sufficient for inhibition of cleavage activity of the Gabija complex.

## Discussion

The Gabija bacterial defense system is a newly discovered bacterial immune defense system contains two components, GajA and GajB. The previous study shows that GajA is essential for bacterial immunity against certain bacteriophages and GajB is also indispensable for Gabija system. GajA is a sequence specific DNA cleaving enzyme and is

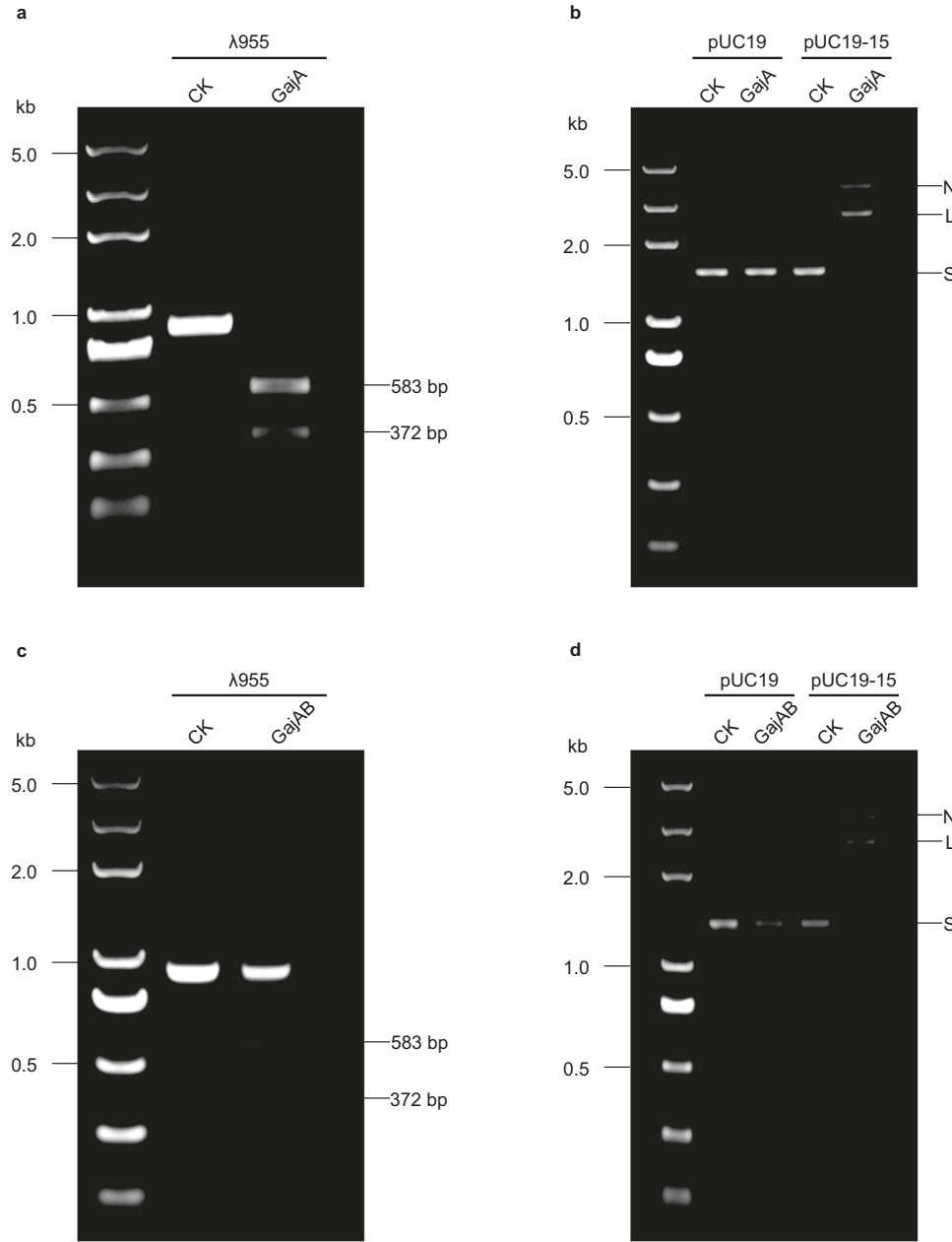

**Fig. 5 | The Gabija complex show substrate preference on supercoiled circular DNA to linear DNA substrate compared with GajA alone. a** Nuclease activity of GajA on λ955 linear DNA substrate; **b** Nuclease activity of GajA on supercoiled plasmids substrate; **c**, Nuclease activity of Gabija complex on λ955 linear DNA substrate; **d** Nuclease activity of Gabija complex on supercoiled plasmids substrate. λ955 is a 955 bp fragment from lambda DNA genome with the recognition sequence (5'-AATAACCCGGATATT-3'). CK, control reaction without GajA or Gabija complex. The pUC19-15 plasmid contains the cleavage recognition sequence (5'-AATAACCCGGATATT-3') and the control is the pUC19 plasmid without the recognition sequence. 'N', 'L', and 'S' represent 'nicked', 'linear', and 'supercoiled' DNA, respectively. All the assays are repeated at least 3 times with similar results. Source data are provided as a Source Data file.

completely inhibited by 0.5 mM ATP. GajB was predicted as a UvrD-like SF1 helicase[3]. Our findings suggest GajB acts as a nucleotide sensor, at least when it is a part of the Gabija complex. Furthermore, our data revealed that with the aid of GajB, the substrate selectivity of the Gabija complex is more stringent, as it prefers supercoiled circular over linear DNA. In addition, compared to GajA, the Gabija complex is more sensitive to the concentration change of ATP. The Gabija complex began to cleave DNA below 2 mM ATP, while GajA began to cleave DNA below 0.5 mM ATP. Moreover, the low concentration (0.05 mM) of ATP promotes the digestion of the linear DNA substrate by the Gabija complex. The ATP concentration is about 3 mM, and the total nucleotide concentration is above 8.7 mM in *E. coli* at physiological conditions[13]. The Gabija complex nuclease activity is inhibited by 2 mM ATP in vitro. The nuclease activity of the Gabija complex should be strictly inhibited at such high physiological concentrations of NTP. Indeed, co-expressing Gabija complex was found not toxic to *E. coli* during the process of purification of the Gabija complex.

Our findings here hence provide insights into the role of the Gabija complex in bacterial anti-phage immune surveillance (Supplementary Fig. 10). The Gabija complex nuclease activity is fully inhibited by nucleotides in the physiological state when bacteria grow normally. During phage invasion, the high-intensity DNA replication and transcription markedly reduced the concentration of nucleotides in the cell. In stage I, the ATP concentration is decreased to (0.5 mM-2 mM ATP) from 3 mM physiological concentration. Therefore, only the cleavage activity of the Gabija complex on supercoiled circular DNA is

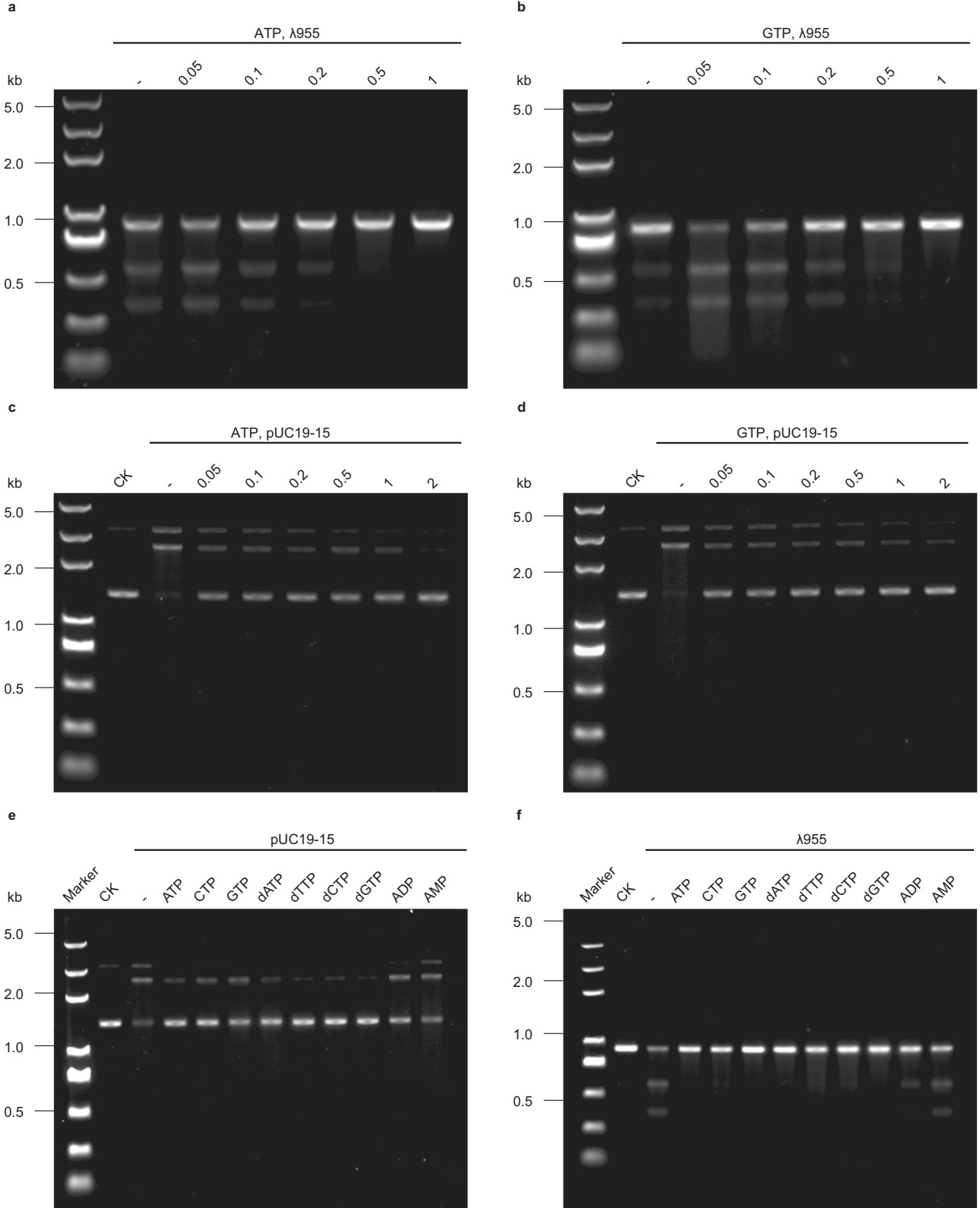

**Fig. 6 | The nuclease activity of Gabija complex is regulated by nucleotides.**
**a** Native agarose gel analysis of nuclease activity of Gabija complex on λ955 linear DNA substrate with the increasing concentration of ATP; **b** Native agarose gel analysis of nuclease activity of Gabija complex on λ955 linear DNA substrate with the increasing concentration of GTP; **c** Native agarose gel analysis of nuclease activity of Gabija complex on supercoiled plasmids substrate pUC19-15 with the increasing concentration of ATP; **d** Native agarose gel analysis of nuclease activity of Gabija complex on supercoiled plasmids substrate pUC19-15 with the increasing concentration of GTP. **e** Effect of NTP, dNTP, ADP, and AMP on nuclease activity of Gabija complex in the presence of pUC19-15 substrate; **f** Effect of NTP, dNTP, ADP, and AMP on nuclease activity of Gabija complex in the presence of λ955 substrate. All the assays are repeated at least 3 times with similar results. Source data are provided as a Source Data file.

activated. In stage II, the ATP concentration is decreased to below 0.5 mM, and the cleavage activity of the Gabija complex on linear DNA is activated. Thus, GajA of the Gabija complex cuts the phage DNA, stopping the infection due to phage replication failure. However, the specific sequence recognition is a double-edged sword. On the one side, it can selectively degrade invading bacteriophages and perform precise DNA substrate selection. Release of occupation of Gabija complex by less effective substrate could improve the efficiency of bacterial immunity. On the other side, bacteriophages would easily mutate away from Gabija defence. It is possible that the palindromic T/A sequences allow formation of a small hairpin in DNA. This would be preferentially formed in supercoiled DNA that is negatively super-coiled, consistent with supercoiled DNA substrate preference of Gabija complex. The recognition of palindrome structure may reduce the possibility of bacteriophage escape due to sequence mutations.

During the revision process, we became aware of three highly relevant preprints and papers that have been posted. The first paper, authored by Antine et al., describes the structural characterization of the Gabija complex[14]. They revealed the structural basis of the Gabija anti-phage defense complex and define a unique mechanism of viral immune evasion by Gad1. The second paper, authored by Cheng et al., uncovers the mechanisms underlying nucleotide sensing by the Gabija complex[15]. GajB senses DNA termini nicked by GajA to hydrolyze (d)A/(d)GTP and results in an efficient abortive infection defense. The third preprint, authored by Fu et al., reports the structural assembly of Gabija complex[16]. These papers, in consistant with our findings reported here, define a detailed framework for understanding the anti-phage immune response by the Gabija complex.

## Methods

### Cloning, expression and purification of Gabija complex

Full-length coding sequence of GajA (residues 1-578) was cloned into PET-52b vectors between XmaI I and BamH I sites harboring a Strep-tag II tag. Full-length coding sequence of GajB (residues 1-493) was cloned into PET-28a vectors between Nde I and Xho I restriction sites with a 6xHis tag. Constructs of full-length GajA and GajB were co-transformed into BL21 (DE3) cells, which were grown at 37 °C in LB medium containing 50 μg/ml kanamycin and 100 μg/ml ampicillin to an OD600 of 0.6-0.7, and then protein expression was induced with 0.5 mM IPTG for 18 h at 16 °C. The cells were collected by centrifugation and resuspended in Buffer A (20 mM Tris-HCl, pH 7.5 at 25 °C, 300 mM NaCl and 2 mM β-mercaptoethanol), and then cells were lysed by sonication. The supernatant after centrifugation was applied onto a Strep-tactin Sepharose column pre-equilibrated in buffer A. After washing with 10-15 column volumes in buffer A, Gabija complex was eluted with buffer A plus 2.5 mM D-desthiobiotin. The elution was concentrated by Millipore Amicon Ultra-15 (30KD MWCO) and further purified by Superdex 200 column in Buffer B (10 mM Tris-HCl pH 7.5, 300 mM NaCl, 2 mM DTT). The peak fractions were collected and kept in Buffer C (50 mM Tris-HCl pH 7.5, 100 mM NaCl, 1 mM DTT, 0.1 mM EDTA, 50% glycerol and 0.1% Triton X-100). GajA and GajB mutants were constructed with site-directed mutagenesis. All the mutants were checked by SDS-PAGE (Supplementary Fig. 11). Sequences of primers are provided in the Supplementary Table 2.

### Size exclusion chromatography coupled to multi-angle light scattering

The purified GajA, GajB and Gabija complex were injected into an Agilent HPLC system with a Superdex 200 10/300 Increase column (GE) at 1 mg/ml in Buffer B at a flow rate of 0.5 ml/min. Eluent from the size exclusion column flowed directly to a static 18-angle light scattering detector (DAWN HELEOS-II) and a refractive index detector (Optilan T-rEX) (Wyatt Technology) with data collected every second. Molar mass was determined using the ASTRA V software. Monomeric

BSA (1 mg/ml) (Sigma) was used for normalization of light scattering detectors and data quality control.

### ATPase/GTPase activity assay

The ATPase/GTPase activity was measured using QuantiChrom ATPase/GTPase assay kit (BioAssay Systems). All reactions were performed in the reaction buffer from the assay kit with various concentrations of ATP/GTP indicated and 0.2 μM protein at 37 °C for 30 min. Then follow the steps of the kit manual and the light absorbance of sample was measured at 620 nm. Nonlinear regression to the Michaelis-Menten equation and statistical analysis were performed using GraphPad Prism 8.

### DNA cleavage assay

The 20 nM various DNA substrate was digested by 0.2 μM protein in a final volume of 10 μl in DNA cleavage buffer (20 mM Tris–HCl, pH 9, 1 mM MgCl2 and 0.1 mg/ml BSA). Reactions were incubated at 37 °C for 5 min and then terminated by the addition of 2 μl of 6×loading buffer containing 60 mM EDTA. Samples were analyzed via 1% native agarose gel electrophoresis and visualized using Bio-Rad ChemiDoc XRS + . After Gelred DNA dye (Sangon biotech) staining, the band of initial DNA substrate was measured and quantified using ImageJ software[17]. The DNA band in protein-free condition lane is taken as intact DNA control to assess the ratio of degraded DNA to intact DNA. The quantification bar graphs represent the average of three independent assays with error bars representing the standard error of the mean.

### Phage resistance assays

Phage cultivation and plaque assays were performed as below[15,18]. The *E. coli* BL21 strain harboring pQE82L vector inserted with various Gabija genes were grown to an OD600 of 0.3 at 37 °C and then protein expression was induced with 0.2 mM IPTG. After cultured for 1 h, 500 μL bacterial cultures were added into 14.5 ml of 0.7% LB top agar and 15 mL mixture were poured onto LB plates containing 100 μg/ml ampicillin and 0.1 mM IPTG. A series of 10-fold dilutions of T7 phage solution was were spotted on double-agar plates. The plates were incubated at 37 °C overnight.

### Cryo-EM grid preparation and date collection

Freshly purified Gabija complex was diluted to 0.5 mg ml$^{-1}$ in 100 mM NaCl using low-salt buffer. The cryo-EM grids were prepared using an automatic plunge freezer Vitrobot Mark IV (ThermoFisher Scientific) under 100% humidity at 4 °C. Each aliquot of 3 μl Gabija complex was loaded onto the freshly glow-discharged homemade Graphene coated Au grid (R1.2/1.3, 300 mesh). The excess sample was blotted with filter paper for 3 s with a blot force of 3, and then the grid was vitrified by flash plunging into liquid ethane.

All datasets were automatically collected using SerialEM-3.7[19] through the beam-image shift data collection method[20] on a 200 kV Talos Arctica microscope (ThermoFisher Scientific) equipped with an energy filter and a K2 Summit detector (Gatan). Images were recorded under the super-resolution mode at a nominal magnification of 130,000× and a defocus value ranged between −1.0 to −3.0 μm, resulting in a pixel size of 1.0 Å per pixel. Each move stack was exposed for 6.4 s with a total exposure dose of ~60 electrons per Å$^2$ over 32 frames.

### Data processing and reconstruction

Each movie stack was first binned and subjected to beam-induced motion correction and anisotropic magnification correction using MotionCor2-1.1.0[21]. The contrast transfer function (CTF) estimation was performed by CTFFIND4.1[22] using non-dose-weighted images. Further processing steps, including particle picking, 2D and 3D classification and auto-refinement were performed in Relion-3.08[23] using images with dose weighting. First, particles were automatic picked

using the Laplacian-of-Gaussian filter and processed with several rounds of reference-free 2D classification. Then, the best five images with clear structural features and different orientations were selected as templates for reference-based particle auto-picking. For Gabija complex, a total of 990,385 particles were picked from 3331 images and subjected to reference-free 2D classification. 615,824 particles with clear structural features were kept for further 3D classification with D2 symmetry, using the initial model generated by Relion-3.08. One among the five classes including 279,580 particles were selected for final 3D auto-refinement, resulting in a 3.6 Å density map. The local resolutions of final maps were calculated by ResMap-1.1.4[24]. Statistics for data collection and processing were summarized in Supplementary Table 1.

## Model building and refinement

For model building, the structure models of GajA and GajB were first predicted by Alphafold 2[25], respectively. Then the complete initial model of Gabija complex was generated by docking the predicted subunit models with a stoichiometric ratio of 4:4 into the reconstituted density map in UCSF Chimera-1.12[26]. After deleting the flexible areas without density map, a few iterative rounds of real-space refinement using PHENIX-1.13[27] and manual modification in Coot-0.8.8[28] were performed until no further improvement could be obtained. Statistics of the final models were summarized in Supplementary Table 1. Cryo-EM single-particle analysis of Gabija complex were summarized in Supplementary Fig. 3. GajA and GajB map to model fit for designated regions is shown in Supplementary Fig. 12. Comparison of Alphafold 2 predicted model and final built model are shown in Supplementary Fig. 13. Figures were prepared in PyMOL and UCSF Chimera-1.12.

## Data availability

The cryo-EM map of Gabija complex has been deposited in the Electron Microscopy Data Bank (EMDB) under the accession code EMD-35977. The structural coordinate of Gabija complex has been deposited in the RCSB Protein Data Bank (PDB) under the accession code 8J4T. Source data are provided with this paper.

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

## Acknowledgements

This research was funded by the National Key R&D Program of China (2022YFA1603702 and 2019YFA0508602 to T.W.) and the National Natural Science Foundation of China (22027810 to Y.H.; 92254304 and 92054108 to T.W.). The authors are indebted to Prof. Xinzheng Zhang (Institute of Biophysics, Chinese Academy of Sciences), Prof. Lixin Wan (Moffit Cancer Center, USA) and Prof. Yu Cao (Shanghai Jiao Tong University, China) for valuble discussions, Dr. Rui Cheng (Huazhong University of Science and Technology, China) for providing phages and for helpful suggestions on phage assays, Fengrui Ren, Jiayun Li and Yining Zhu (Institute of Biophysics, Chinese Academy of Sciences) for technical assistance, and Xiaoxia Yu for support with SEC-MALS. Cryo-EM data collection was carried out at the Center for Biological Imaging (CBI), Core Facilities for Protein Science at the Institute of Biophysics, Chinese Academy of Sciences. We thank Boling Zhu, Lihong Chen, Xujing Li, Xiaojun Huang and other staff members at the CBI for their support with data collection.

## Author contributions

Y.H., Y.Z., M.X., K.D. and S.X. performed protein purification and biochemical experiments. J.M. and L.K. collected cryo-EM data and determined structure. X.Y. contributed to discussions. T.W. supervised experiments. Y.H. drafted the manuscript with input from all the others. All authors contributed to data analysis and manuscript preparation.

## Competing interests

The authors declare no competing interests.
