## [Peer Review File · Nature Communications]

Structural and biochemical insights into the mechanism of the Gabija bacterial immunity systemREVIEWER COMMENTS

Reviewer #1 (Remarks to the Author):

Structural and biochemical insights into the mechanism of the Gabija bacterial immunity system

Huo et al.

I was asked to review the cryo-EM technical aspects of this manuscript, something I find myself unable to do due to the authors' refusal to abide by the data transparency standards in the field.

This began with the initial submission. Surprisingly, a manuscript whose main finding was a cryo-EM structure of the Gabija complex was virtually devoid of cryo-EM data. Other than an incomplete cryo-EM table, the manuscript had no figures, either main or supplementary, showing their cryo-EM map, data processing, model fit, etc. I wrote to the editor to point this out and to say I would look at the manuscript once that data was available.

A week later, the editor let me know that a supplementary figure on cryo-EM data processing, and a PDB Validation Report had now been submitted. That turned out not to be entirely true. While there was a supplementary figure outlining the cryo-EM data processing, it is still inadequate. The FSC curve does not show all the standard curves, and there is no indication (visual or quantitative) of how well the model fits the map. Worse, the PDB validation report is clearly labeled as "NOT for manuscript review", because the authors had deposited neither their PDB model, nor the raw full and half cryo-EM maps needed for resolution estimation and map-to-model fit calculations. As clearly stated in the report, it was generated by the authors using the validation server. I wrote to the editor again, letting her know I would not review the paper until the maps and models had been deposited in the EMDB and PDB, and a new validation report, fit for review, was generated.

A few days later, the editor sent me a link to a folder with cryo-EM data: filtered map, raw half maps, and model. However, as far as I can tell the authors have not yet deposited these data in the PDB and EMDB, and thus still lack a PDB Validation Report. Furthermore, even the data provided in that folder appears incorrect. The unfiltered half maps look strongly filtered and are visually clearly of much lower resolution than the final filtered map. This makes no sense as it is the opposite of what one would expect: the half maps should be noisier (i.e. look more "detailed") than the filtered map.

The authors must deposit their data and include PDB and EMDB entry numbers in their manuscript, along with a PDB Validation Report fit for review. More cryo-EM data is needed in the manuscript as well to let readers judge the structure without having to obtain maps and analyze them themselves. If the authors refuse to abide by current standards in scientific publishing, I am unwilling to waste my time reviewing this manuscript. We are no longer in the 1990s, when structures were routinely published with no data made available.

Reviewer #2 (Remarks to the Author):

This paper reports the structure (by cryoEM) of the Gabija protein complex for antiphage defence, consisting of an octameric ring with 4 copies of each subunit. This is the first structural data for the Gabija complex, revealing key new mechanistic and structural details. Only 2 of the four domains of the SF1 helicase GajB are observed, which the authors ascribe to structural flexibility. The structural work appears carefully done, but should be reviewed by a cryoEM expert.

The authors proceed to investigate the activity of the complex with respect to substrate specificity and sensitivity to NTP/dNTP levels. GajA was previously shown to be a DNA nickase with DNA sequence specificity, cutting sites with a CCGG core sequence (Cheng et al. 2021). Here, the authors first demonstrate that mutations of the basic patch around the active site of GajA can abolish nuclease activity (Figure 4). They go on to demonstrate that the Gabija complex prefers supercoiled over linear substrates (Figure 5), and investigate the effect of nucleotides on the activity (figure 6).

Overall, the new structural information and the analysis of the activity of the GajAB complex represent a significant step forwards in understanding. However, the work leaves many questions unanswered, including the role of the GajB subunit and the requirement for 2 ATPase active sites in the complex. ATP binding and hydrolysis by SF1 helicases results in cyclical conformational changes linked to the manipulation and unwinding of DNA, but this is not explored or even really touched on in the manuscript.

Major points:

1. In figure 4, all three basic patch variants affect the nuclease activity, with mut2 and mut3 abolishing activity and mut1 resulting in loss of specificity. Can the authors explain these observations more fully? At present the analysis is minimal (bottom of page 8).
2. The extended recognition sequence of the GajAB complex is intriguing, as it seems counterintuitive to impose such a filter on a defence system. Viruses would easily mutate away from Gabija defence. Can the authors reflect on this in the discussion?

3. Both GajA and GajB are ATPases in vitro – activities that are not really understood in the context of the proposed model of inhibition at high ATP levels – why hydrolyse ATP at all? What effect does non-hydrolysable analog of ATP have on activity? (eg ATPγS or AMP-PNP). This would be easy to test and could show whether binding is sufficient for inhibition or conversely that ATP hydrolysis is required.
4. In Ext data fig6, the most striking phenotype observed is that the ABmut enzyme is much more active in degrading linear DNA than the wt protein. This observation is not described or explained – how do the authors interpret this?

Minor points

1. In figure 1, it is hard for the reader to immediately recognise which subunits are GajA and which GajB. Perhaps add a figure part coloured by subunit identity?
2. P4. Avoid phrases such as “we further proved” – claims to have proven something should not be stated – we generate data that support (or disprove) hypotheses.
3. P10 typo “circular?”
4. P37 ext data fig6 – panel f. mentioned instead of e.

Reviewer #3 (Remarks to the Author):

Gabija is a recently identified bacterial immune antiviral defense system that consists of GajA and GajB proteins. The exact mechanism of Gabija immune function is currently unknown however it is evident from prior published data that both GajA and GajB are required. In this manuscript, Huo and colleagues provide new structural and functional data to support a mechanism of GajA nuclease activity being regulated through direct protein-protein interaction with GajB and build on prior work to reveal a marked dependence on intracellular nucleotide concentration. The authors provide the first reported cryo-EM structure of the complex formed between GajA and GajB revealing a 4:4 ratio of subunits. Evidence is also provided that the Gabija complex is potently activated (or rather disinhibited) to cleave circular DNA by low ATP concentration and inhibited at higher levels. There is a high level of interest in the field for structural and biochemical characterization of this newly defined defense system and as such these data are valuable. The manuscript in its current form does provide an intriguing model for how free-nucleotide depletion during phage replication could lead to activation of the Gabija system. However, the data and figures are presented in a way that is inconsistent with the standards and expectations of the field and separately, some of the main hypotheses generated in this manuscript will likely need to be tested by experiments with phage. Outlined below are suggestions and comments that may aid in productive revision of the manuscript before resubmission for peer-review.

Concerns:

- Abstract line 1: The word “predicted” here does not apply any longer as published evidence for immune activity of this system is cited by the authors.
- Introduction, second paragraph: “Sorek et al. predicted computationally and validated experimentally 10 previously unknown bacterial immune defense systems^{3,4}.” The reference #3 to the 2018 Sorek et al. manuscript (should read Doron et al.) is correct but reference #4 has no relevance to this manuscript nor to the sentence for which the citation is attached and should therefore be removed.
- References to many relevant and more recent work in the field of prokaryotic innate immune systems is lacking in this manuscript. In fact, only 8 references to non-software/cryo-EM methods papers are cited which feels very limited. It should be mentioned in greater detail that Gabija is one of now 100s of newly identified immune defense mechanisms in prokaryotes and appropriate references cited.
- Introduction: “We further proved that the Gabija complex has the sequence-specific DNA endonuclease activity and prefers supercoiled circular DNA as its substrate.” The word “proved” is too strong/absolute. Certainly, evidence has been provided that suggests the Gabija complex has DNA sequence preference and prefers supercoiled circular DNA. However, this has not yet been shown in vivo.
- Generally, the color choices in the structure figures are inconsistent between panels and not beneficial to the reader. The abundance of colors used is overwhelming and contrast is not used appropriately.
- In Figure 1, labeling of the individual subunits A-G is less useful than highlighting GajA vs GajB. Representing the hetero-octameric structure could be done more clearly by showing one heterodimer in colors and the other three in grey (for example).
- In Figure 4A, showing hydrogen atoms on all the lysine residues is not useful or accurate especially given the resolution of the cryo-EM data and additionally Figure 4B is somewhat redundant with 4A. The authors could choose to just show 4B alone as there is more information content.
- Page 4- further explanation is needed for “ATPase domain of GajA adopts a productive conformation.”
- Page 7 and Figure 3- What is the RMSD value for the structural alignment of the structures (GajA and OLD)?
- Pages 8 and 11- The authors did not mention how the mutants were created in the methods section. Additionally, authors did not confirm/indicate if the mutants were properly folded.
- Extended Data Figures 4 and 6- The authors must show individual data points not just the average and error bars. The statistical analysis used to generate the error bars need to be explained in the figure legend (SEM? SD?)
- Extended Data Figures 4 and 6- Why are these activity plots separated out into two separate figures? The GTPase level of GajB appears much weaker than the ATPase rate. Can the authors comment more on this subject? Possibly also testing GTPase activity for the GajA and GajAB complex?
- “The octameric ring formation can be best described as three steps: protomer formation of GajA-GajB heterodimer and two sequential dimerization.” In this description the authors make it appear that they

know the order of steps leading to Gabija complex formation. The language should be modified to reflect that this is hypothetical.

- What does “The cleavage activity of the Gabija complex on supercoiled plasmid is not affected at all” mean? Was this a comparison to the GajA protein alone?
- “This substrate selectivity is more efficient in bacterial immune response...” needs to be explicitly tested/shown if this phrase is to remain in the manuscript. Indeed, the following sentence needs to be justified through experiment as well.
- In the discussion section- “Although GajB was predicted as a UvrD-like SF1 helicase” what is the intent behind use of the word “although” in this context?
- The description of the model for how GajAB is activated by nucleotide depletion during phage infection and its overall function in immunity could benefit from a figure panel to help guide readers.
- As Alphafold2 models for GajA and GajB were docked into the maps that were obtained, it would also be useful to present/discuss how different the models were relative to the final build. Were the conformations differing significantly from what was predicted? It is mentioned that some “density” was missing such that the full structures were not able to be built into the map. It might help to show this analysis in an extended data figure.
- Page 4- define OLD – overcoming lysogenization defect.
- Page 4- define Toprim – Topoisomerase/Primase.
- Page 5- define CTR and NTR – C-terminal region and N-terminal region.
- Page 10- delete the question mark after “circular.”
- Page 16- delete extra space between “beta-mercaptoethanol” and the colon.

Point-by-point Response to Reviewers' Comments

Reviewer #1 (Remarks to the Author):

Structural and biochemical insights into the mechanism of the Gabija bacterial immunity system
Huo et al.

I was asked to review the cryo-EM technical aspects of this manuscript, something I find myself unable to do due to the authors' refusal to abide by the data transparency standards in the field. This began with the initial submission. Surprisingly, a manuscript whose main finding was a cryo-EM structure of the Gabija complex was virtually devoid of cryo-EM data. Other than an incomplete cryo-EM table, the manuscript had no figures, either main or supplementary, showing their cryo-EM map, data processing, model fit, etc. I wrote to the editor to point this out and to say I would look at the manuscript once that data was available.

A week later, the editor let me know that a supplementary figure on cryo-EM data processing, and a PDB Validation Report had now been submitted. That turned out not to be entirely true. While there was a supplementary figure outlining the cryo-EM data processing, it is still inadequate. The FSC curve does not show all the standard curves, and there is no indication (visual or quantitative) of how well the model fits the map. Worse, the PDB validation report is clearly labeled as "NOT for manuscript review", because the authors had deposited neither their PDB model, nor the raw full and half cryo-EM maps needed for resolution estimation and map-to-model fit calculations. As clearly stated in the report, it was generated by the authors using the validation server. I wrote to the editor again, letting her know I would not review the paper until the maps and models had been deposited in the EMDB and PDB, and a new validation report, fit for review, was generated. A few days later, the editor sent me a link to a folder with cryo-EM data: filtered map, raw half maps, and model. However, as far as I can tell the authors have not yet deposited these data in the PDB and EMDB, and thus still lack a PDB Validation Report. Furthermore, even the data provided in that folder appears incorrect. The unfiltered half maps look strongly filtered and are visually clearly of much lower resolution than the final filtered map. This makes no sense as it is the opposite of what one would expect: the half maps should be noisier (i.e. look more "detailed") than the filtered map.

The authors must deposit their data and include PDB and EMDB entry numbers in their manuscript, along with a PDB Validation Report fit for review. More cryo-EM data is needed in the manuscript as well to let readers judge the structure without having to obtain maps and analyze them themselves. If the authors refuse to abide by current standards in scientific publishing, I am unwilling to waste my time reviewing this manuscript. We are no longer in the 1990s, when structures were routinely published with no data made available.

Response: We are very grateful for your objective and professional comments on the cryo-EM technical aspects of the manuscript. We deeply regret the unprofessional manner in which we handled the structural data in the previous version of our manuscript, causing confusion for reviewers and inconvenience to the editor. We genuinely apologize for any frustration this may have caused.

Initial Submission: We acknowledge that our initial submission lacked essential cryo-EM data, which was an oversight on our part. We apologize for any inconvenience this may have caused. It is important to clarify that we have never refused to adhere to current standards in scientific publishing. Upon receiving the request from the editor, we promptly took the necessary steps to rectify the situation. Specifically, we deposited the cryo-EM density map and the related model into the EMDB and PDB databases. Additionally, we provided the cryo-EM data as a supplementary file along with the cryo-EM density map and related model. We understand your concern regarding the delay in providing a formal validation report for manuscript review. The time required for its generation was beyond our control and currently more than 7-10 days (26 days in our case). We have submitted the formal validation report after the initial submission.

Supplementary Data: We acknowledge the insufficiency of the supplementary figure on cryo-EM data processing. Your comments regarding the FSC curve and the absence of quantitative information on model fit are duly noted. We are committed to providing a comprehensive and transparent account of our cryo-EM data and will ensure that these aspects are properly represented in our revised manuscript.

Cryo-EM Data Folder: We appreciate the feedback regarding the cryo-EM data provided in the folder. We have carefully reviewed the data to ensure the accuracy and confirmed that the half-maps provided earlier were directly generated by the software Relion without any other processing. Additionally, we will make sure to deposit these data in the PDB and EMDB as you have rightly suggested.

Incorporating More Cryo-EM Data: We agree with your suggestion to include more cryo-EM data in the manuscript to facilitate readers' understanding and evaluation of the structure. We are committed to providing a more comprehensive representation of our cryo-EM findings, allowing for a thorough assessment without requiring readers to obtain and analyze maps independently. Specifically, we added more information in Extended Data Fig.3, Extended Data Fig.4, Extended Data Fig.12 and Extended Data Table 1. To ensure transparency and accessibility, we have deposited the cryo-EM density map and related model coordinate in the databases EMDB and PDB with accession codes EMD-35977 and 8J4T, respectively.

Adherence to Scientific Publishing Standards: We genuinely value the importance of adhering to current standards in scientific publishing. We apologize for any perception that we may not be in line with these standards, and we assure you that we are fully committed to upholding transparency and rigor in our research.

We understand the importance of your expertise and time commitment as a reviewer. We sincerely apologize for any inconvenience our previous submission may have caused and assure you that we are actively working to address all the concerns you have raised. We believe that the revisions made to our manuscript, along with the availability of the cryo-EM data and validation report, will address your concerns and contribute to a more robust and transparent presentation of our research. Additionally, we want to emphasize our commitment to providing any further necessary data and revisions promptly during the ongoing review process.

Reviewer #2 (Remarks to the Author):

This paper reports the structure (by cryoEM) of the Gabija protein complex for antiphage defence, consisting of an octameric ring with 4 copies of each subunit. This is the first structural data for the Gabija complex, revealing key new mechanistic and structural details. Only 2 of the four domains of the SF1 helicase GajB are observed, which the authors ascribe to structural flexibility. The structural work appears carefully done, but should be reviewed by a cryoEM expert.

Response: We thank the reviewer for the constructive comments which have been very instrumental in guiding us further strengthening the manuscript as detailed in the point-by-point responses below.

The authors proceed to investigate the activity of the complex with respect to substrate *specificity* and sensitivity to NTP/dNTP levels. GajA was previously shown to be a DNA nickase with DNA sequence specificity, cutting sites with a CCGG core sequence (Cheng et al. 2021). Here, the authors first demonstrate that mutations of the basic patch around the active site of GajA can abolish nuclease activity (Figure 4). They go on to demonstrate that the Gabija complex prefers supercoiled over linear substrates (Figure 5), and investigate the effect of nucleotides on the activity (figure 6). Overall, the new structural information and the analysis of the activity of the GajAB complex represent a significant step forward in understanding. However, the work leaves many questions unanswered, including the role of the GajB subunit and the requirement for 2 ATPase active sites in the complex. ATP binding and hydrolysis by SF1 helicases results in cyclical conformational changes linked to the manipulation and unwinding of DNA, but this is not explored or even really touched on in the manuscript.

Response: We thank the reviewer for the insightful comment. The structures under different conformational states will be difficult to obtain. We have attempted to solve the structure of GajAB system in complex with DNA and nucleotide analogue AMP-PNP. But the density of DNA and nucleotide analogue AMP-PNP could not be traced in the electron density map, though they are included in the cryo-EM sample. However, we will continue to make efforts in future studies on the questions you raised, to comprehensively and completely elucidate the molecular mechanism of GajAB system.

Major points:

1. In figure 4, all three basic patch variants affect the nuclease activity, with mut2 and mut3 abolishing activity and mut1 resulting in loss of specificity. Can the authors explain these observations more fully? At present the analysis is minimal (bottom of page 8).

Response: Due to the lack of structural information of GajAB complex with DNA substrates, it could only be inferred from biological mutation experiments, the selection of sequence specific DNA

digestion of GajA may be related to interactions between the phosphate backbone of the DNA substrate palindrome structure and GajA_{mut1} region.

2. The extended recognition sequence of the GajAB complex is intriguing, as it seems counterintuitive to impose such a filter on a defence system. Viruses would easily mutate away from Gabija defence. Can the authors reflect on this in the discussion?

Response: Thanks for your helpful comment. Specific sequence recognition is a double-edged sword. On the one side, it can selectively degrade invading bacteriophages and perform precise DNA substrate selection. Release of occupation of Gabija complex by less effective substrate could improve the efficiency of bacterial immunity. On the other side, bacteriophages would easily mutate away from Gabija defence.

3. Both GajA and GajB are ATPases in vitro – activities that are not really understood in the context of the proposed model of inhibition at high ATP levels – why hydrolyse ATP at all? What effect does non-hydrolysable analog of ATP have on activity? (eg ATPγS or AMP-PNP). This would be easy to test and could show whether binding is sufficient for inhibition or conversely that ATP hydrolysis is required.

Response: We measure the effect of non-hydrolysable analog of ATP on the cleavage activity of the Gabija complex (Extended Data Fig.9). The results reveal that the cleavage activity of the Gabija complex is also inhibited by AMP-PNP. This indicates the binding of ATP is sufficient for inhibition of cleavage activity of the Gabija complex.

4. In Ext data fig6, the most striking phenotype observed is that the AB_{mut} enzyme is much more active in degrading linear DNA than the wt protein. This observation is not described or explained – how do the authors interpret this?

Response: We thank the reviewer for recognizing this exciting finding. This striking phenotype observed is that the AB_{mut} Gabija complex is much more active in degrading linear DNA than the wild-type protein (Extended Data Fig.7g of the revised manuscript). This result further revealed that the ATPase activity of GajB is related to the substrate selectivity of the Gabija complex, making Gabija complex prefer supercoiled circular over linear DNA.

Minor points

1. In figure 1, it is hard for the reader to immediately recognise which subunits are GajA and which GajB. Perhaps add a figure part coloured by subunit identity?

Response: To better visualize each subunit of the GajAB complex, Figure 1 was re-generated and the structures are now colored by subunit identity.

2. P4. Avoid phrases such as “we further proved” – claims to have proven something should not be stated – we generate data that support (or disprove) hypotheses.

Response: As suggested, we have revised “We further proved” to “Our data further suggest”.

3. P10 typo “circular?”

Response: We apologize sincerely for the typo. we have corrected this typo in the revised manuscript. The whole sentence is revised to “The cleavage activity of the Gabija complex on linear DNA substrate decreased drastically as only about 30% of the linear DNA substrate is digested by Gabija complex.”

4. P37 ext data fig6 – panel f. mentioned instead of e.

Response: We thank the reviewer for this good catch. We have revised the panel f into panel e.

Reviewer #3 (Remarks to the Author):

Gabija is a recently identified bacterial immune antiviral defense system that consists of GajA and GajB proteins. The exact mechanism of Gabija immune function is currently unknown however it is evident from prior published data that both GajA and GajB are required. In this manuscript, Huo and colleagues provide new structural and functional data to support a mechanism of GajA nuclease activity being regulated through direct protein-protein interaction with GajB and build on prior work to reveal a marked dependence on intracellular nucleotide concentration. The authors provide the first reported cryo-EM structure of the complex formed between GajA and GajB revealing a 4:4 ratio of subunits. Evidence is also provided that the Gabija complex is potentially activated (or rather disinhibited) to cleave circular DNA by low ATP concentration and inhibited at higher levels. There is a high level of interest in the field for structural and biochemical characterization of this newly defined defense system and as such these data are valuable. The manuscript in its current form does provide an intriguing model for how free-nucleotide depletion during phage replication could lead to activation of the Gabija system. However, the data and figures are presented in a way that is inconsistent with the standards and expectations of the field and separately, some of the main hypotheses generated in this manuscript will likely need to be tested by experiments with phage. Outlined below are suggestions and comments that may aid in productive revision of the manuscript before resubmission for peer-review.

Concerns:

- Abstract line 1: The word “predicted” here does not apply any longer as published evidence for immune activity of this system is cited by the authors.

Response: We have revised the word “predicted” to “discovered”.

- Introduction, second paragraph: “Sorek et al. predicted computationally and validated experimentally 10 previously unknown bacterial immune defense systems^{3,4}.” The reference #3 to the 2018 Sorek et al. manuscript (should read Doron et al.) is correct but reference #4 has no relevance to this manuscript nor to the sentence for which the citation is attached and should therefore be removed.

Response: We thank the reviewer for picking up this error. We have revised the “Sorek et al.” to “Doron et al.”, and removed the reference #4.

- References to many relevant and more recent work in the field of prokaryotic innate immune systems is lacking in this manuscript. In fact, only 8 references to non-software/cryo-EM methods papers are cited which feels very limited. It should be mentioned in greater detail that Gabija is one of now 100s of newly identified immune defense mechanisms in prokaryotes and appropriate references cited.

Response: As per the reviewer's suggestion, we have included the new references #4, #5, #6, #7, #8, #14, #15, #17, #27.

- Introduction: "We further proved that the Gabija complex has the sequence-specific DNA endonuclease activity and prefers supercoiled circular DNA as its substrate." The word "proved" is too strong/absolute. Certainly, evidence has been provided that suggests the Gabija complex has DNA sequence preference and prefers supercoiled circular DNA. However, this has not yet been shown *in vivo*.

Response: We have changed "we further proved" to "Our data further suggest"; We also conducted phage defense assay of Gabija complex *in vivo* (Extended Data Fig.2). Both GajA and GajB are essential for anti-phage activity of Gabija system; deletion of either gene leads to the loss of Gabija immunity revealed by phage resistance assay *in vivo*. The mutations in two ATPase active sites of Gabija complex both abolish the anti-phage activity of the Gabija system. The most striking phenotype observed is that the AB_{mut} Gabija complex is much more active in degrading linear DNA than the wild-type protein (Extended Data Fig.7g). This result further revealed that the ATPase activity of GajB is related to the substrate selectivity of the Gabija complex, making Gabija complex prefer supercoiled circular over linear DNA. The loss of circular DNA substrate preference in AB_{mut} Gabija complex may result in abolishing the phage resistance of Gabija system *in vivo* (Extended Data Fig.2).

- Generally, the color choices in the structure figures are inconsistent between panels and not beneficial to the reader. The abundance of colors used is overwhelming and contrast is not used appropriately.

- In Figure 1, labeling of the individual subunits A-G is less useful than highlighting GajA vs GajB. Representing the hetero-octameric structure could be done more clearly by showing one heterodimer in colors and the other three in grey (for example).

Response: To better visualize each subunit of the GajAB complex, Figure 1 was re-generated and the structures are now colored by subunit identity.

- In Figure 4A, showing hydrogen atoms on all the lysine residues is not useful or accurate especially given the resolution of the cryo-EM data and additionally Figure 4B is somewhat redundant with 4A. The authors could choose to just show 4B alone as there is more information content.

Response: As suggested, Figure 4A has been removed from the revised manuscript.

- Page 4- further explanation is needed for "ATPase domain of GajA adopts a productive conformation."

Response: In the revised manuscript, explanation was added: The ATPase domain of GajA encodes

a conserved active site and adopts a productive conformation, with the ABC signature sequence orienting toward the active site of the opposing subunit in close approach to the P loop and bound nucleotide.

- Page 7 and Figure 3- What is the RMSD value for the structural alignment of the structures (GajA and OLD)?

Response: As requested, we added the RMSD value for the structural alignment of the structures (GajA and OLD). The RMSD of structural alignment of GajA Toprim domains from *Bacillus cereus* VD045 (orange) and *Burkholderia pseudomallei* (OLD_Bp, PDB: 6NK8; purple) is 3.2 Å.

- Pages 8 and 11- The authors did not mention how the mutants were created in the methods section. Additionally, authors did not confirm/indicate if the mutants were properly folded.

Response: As per the reviewer's suggestion, GajA and GajB mutants were constructed with site-directed mutagenesis. All the mutants were checked by SDS-PAGE (Extended Data Fig. 11).

- Extended Data Figures 4 and 6- The authors must show individual data points not just the average and error bars. The statistical analysis used to generate the error bars need to be explained in the figure legend (SEM? SD?)

Response: As instructed, the individual data points of three independent experiments are indicated in the Extended Data Fig.14. The error bars of the statistical analysis in the figure legend are indicated by SD.

- Extended Data Figures 4 and 6- Why are these activity plots separated out into two separate figures? The GTPase level of GajB appears much weaker than the ATPase rate. Can the authors comment more on this subject? Possibly also testing GTPase activity for the GajA and GajAB complex?

Response: Again, we thank the reviewer for this comment. We merged the Extended Data Figures 4 and 6 as the revised Extended Data Figures 7.

The GTPase level of GajB appears much weaker than the ATPase rate is caused by the substrate selectivity. The Q motif is supposed to be situated in the $\alpha 1$ helix of the GajB structure, while the Q motif is missing in $\alpha 1$ helix. The adenine base is selected explicitly by Q motif, the missing of the Q motif results in the loss of selectivity of ATP. The GTPase activity of GajB is observed indeed in the following biochemical assay, but the GTPase activity of GajB appears much weaker than the ATPase rate, which indicating that the ATPase domain of GajB still has a higher affinity for adenine than guanosine. (Extended Data Fig. 7b and 7e).

We have tested GTPase activity for the GajA and GajAB complex following the suggestion. As shown

in the paper: We also observed that the V_{max} of GajA, GajB and Gabija complex activity of GTP hydrolysis is 1.364 mM/min, 0.218mM/min and 1.829 mM/min, respectively (Extended Data Fig. 7d, 7e, 7f). In the case of GTP substrate, there is also a positive synergistic effect, although it is weaker than ATP substrate, which is consistent with the affinity of the substrate.

- “The octameric ring formation can be best described as three steps: protomer formation of GajA-GajB heterodimer and two sequential dimerization.” In this description the authors make it appear that they know the order of steps leading to Gabija complex formation. The language should be modified to reflect that this is hypothetical.

Response: We apologize sincerely for the misleading description. We have redefined the interactions between them based on the SEC-MALS results as detailed below.

SEC-MALS results show that individual GajA and GajB exist in the form of tetramer and monomer respectively (Extended Data Fig.1c,1d), supporting hypothesis that the octameric ring of Gabija complex is formed through two steps: the tetramer formation of GajA and subsequent octamer formation of Gabija complex. The tetramer formation of GajA leads to the formation of interactions between GajA A1-A2 and A1-A3 (interface A1-A2, Fig.1c; interface A1-A3, Fig.1d;). The A1-A2 interface involves symmetric interactions, the contact sites are mainly related to ATPase domain of GajA. They are $\alpha 2$, $\beta 6$, the loop between $\alpha 5$ and $\beta 8$, the loops on both sides of the $\beta 6$ (Fig.1c). The A1-A3 interface involves symmetric interactions. These contacting sites spread both the ATPase and Toprim domains of GajA. They are $\alpha 5$, $\alpha 6$, $\alpha 7$, $\alpha 8$, $\alpha 10$, $\alpha 15$, $\alpha 16$, $\beta 13$, the loop between $\alpha 14$ and $\alpha 15$, the loop between $\alpha 10$ and $\beta 14$, the loop between $\alpha 8$ and $\beta 12$, the loops on both sides of the $\beta 13$, the loop between $\alpha 7$ and $\beta 11$, the loop between $\beta 9$ and $\beta 10$, the loop between $\alpha 1$ and $\beta 3$, the loop between $\alpha 6$ and $\beta 8$, (Fig.1d). The octamer formation of Gabija complex leads to the formation of interactions between GajA1-GajB1, GajA1-GajB2 and GajB1-GajB2; (interface A1-B1, Fig.1e; interface A1-B2, B1-B2; Fig.1f;). The interactions within GajA1-GajB1 are mediated by the ATPase domain of GajA and the 1A, 1B domains of GajB. The $\alpha 3$ helix, the loops on both sides of the $\alpha 3$ helix, the loop between $\alpha 5$ and $\beta 8$ and the $\alpha 4$ helix of GajA are involved in this contact interface. In GajB, the interface contact sites are $\alpha 4$, $\alpha 7$, the loop between $\alpha 2$ and $\beta 2$ and the loop between $\alpha 3$ and $\beta 3$ (Fig.1e). The flexible CTR region (C-terminal region) (residues 223-416) of GajB could not be traced in the electron density map; To determine the interactions between GajA and CTR region of GajB, two constructs of GajB have been made, the NTR region (N-terminal region) (residues 1-223) and the CTR region (residues 223-416). Size exclusion chromatography demonstrates that the NTR region of GajB indeed binds to GajA, while the CTR region of GajB doesn't bind to GajA (Extended Data Fig.5a, 5b). Interface A1-B2 is mediated by the $\beta 5$, $\beta 6$, the loop between $\beta 5$ and $\beta 6$ of GajA ATPase domain and by the loop between $\alpha 3$ and $\beta 3$ of GajB 1A domain. Interface B1-B2 is mediated by the $\alpha 5$ and the loops on both sides of the $\alpha 5$ (Fig.1f).

- What does “The cleavage activity of the Gabija complex on supercoiled plasmid is not affected at all” mean? Was this a comparison to the GajA protein alone?

Response: Yes, this is a comparison to the GajA protein alone. The cleavage activity of the Gabija complex on the supercoiled plasmid is not affected at all compared to GajA protein alone.

- “This substrate selectivity is more efficient in bacterial immune response...” needs to be explicitly tested/shown if this phrase is to remain in the manuscript. Indeed, the following sentence needs to be justified through experiment as well.

Response: We have deleted these speculations.

- In the discussion section- “Although GajB was predicted as a UvrD-like SF1 helicase” what is the intent behind use of the word “although” in this context?

Response: We have deleted the word “although”.

- The description of the model for how GajAB is activated by nucleotide depletion during phage infection and its overall function in immunity could benefit from a figure panel to help guide readers.

Response: As suggested, we have added a schematic diagram to illustrate the mechanism of the Gabija defense system in Extended Data Fig.10 to help future readers to better understand our works.

- As AlphaFold2 models for GajA and GajB were docked into the maps that were obtained, it would also be useful to present/discuss how different the models were relative to the final build. Were the conformations differing significantly from what was predicted? It is mentioned that some “density” was missing such that the full structures were not able to be built into the map. It might help to show this analysis in an extended data figure.

Response: As per the reviewer’s suggestion, we have described the difference between the AlphaFold 2 predicted model and final built model in the Method section and shown it in Extended Data Fig.13. The residues 157-258 of GajA and 224-494 of GajB in the AlphaFold 2 predicted model is missing the final built model.

- Page 4- define OLD – overcoming lysogenization defect.

Response: We have defined the OLD (overcoming lysogenization defect) in the revised manuscript.

- Page 4- define Toprim – Topoisomerase/Primase.

Response: We have defined the Toprim (Topoisomerase/Primase) in the revised manuscript.

- Page 5- define CTR and NTR – C-terminal region and N-terminal region.

Response: We have defined CTR (C-terminal region) and NTR (N-terminal region) in the revised manuscript.

- Page 10- delete the question mark after “circular.”

Response: We have changed the typo “circular” into “linear”. The revised sentence now reads: “The cleavage activity of the Gabija complex on linear DNA substrate decreased drastically as only about 30% of the linear DNA substrate is digested by Gabija complex.”

- Page 16- delete extra space between “beta-mercaptoethanol” and the colon.

Response: As suggested, we have deleted extra space between “beta-mercaptoethanol” and the bracket.

REVIEWERS' COMMENTS

Reviewer #1 (Remarks to the Author):

I am satisfied with the changes the authors have made to the manuscript to include a reasonable amount of cryo-EM data for readers to judge the quality of the structures. Maps and models have been deposited and PDB Validation reports for reviewers are now included in the re-submission.

Reviewer #2 (Remarks to the Author):

In this revised manuscript, the authors have responded to my queries and provided further data on non-hydrolysable analogues of ATP.

Regarding point 2, the extended recognition sequence, the authors' response doesn't really add much to the analysis. Thinking about the recognition site, it is possible that the palindromic T/A sequences allow formation of a small hairpin in DNA. This would be preferentially formed in supercoiled DNA that is negatively supercoiled, which would fit the data presented in the paper. There are ways to test this, for example by making a fixed four-way junction in linear DNA using the sequence. However, given this is review stage 2 and there are other papers in press it might be best just to mention the possibility in the discussion.

Reviewer #3 (Remarks to the Author):

This reviewer is mostly satisfied by the substantial new experimental data and analysis provided by the authors in this revised version of the manuscript. Additional evidence for the folding and enzymatic activity of Gabija is now included that strengthens the original draft.

The plaque assay presented in Extended Data Figure 2 appears possibly overloaded and 7-8 logs of protection is almost unbelievable. Trusting that all sample lanes were loaded the same, the phenotype is remarkable. A minor concern is that there is limited utility of the color scheme used in Extended Data Figure 8. Consider another version that may rely less on readers ability to distinguish between closely related or otherwise complementary colors.

REVIEWERS' COMMENTS

Reviewer #1 (Remarks to the Author):

I am satisfied with the changes the authors have made to the manuscript to include a reasonable amount of cryo-EM data for readers to judge the quality of the structures. Maps and models have been deposited and PDB Validation reports for reviewers are now included in the re-submission.

Response: We thank the reviewer for the constructive comments, which have been very instrumental in guiding us to strengthen the manuscript.

Reviewer #2 (Remarks to the Author):

In this revised manuscript, the authors have responded to my queries and provided further data on non-hydrolysable analogues of ATP.

Regarding point 2, the extended recognition sequence, the authors' response doesn't really add much to the analysis. Thinking about the recognition site, it is possible that the palindromic T/A sequences allow formation of a small hairpin in DNA. This would be preferentially formed in supercoiled DNA that is negatively supercoiled, which would fit the data presented in the paper. There are ways to test this, for example by making a fixed four-way junction in linear DNA using the sequence. However, given this is review stage 2 and there are other papers in press it might be best just to mention the possibility in the discussion.

Response: Thank you very much for the constructive comments. We add this possibility in the discussion of the revised manuscript: **It is possible that the palindromic T/A sequences allow formation of a small DNA hairpin. This would be preferentially formed in supercoiled DNA that is negatively supercoiled, consistent with supercoiled DNA substrate preference of Gabija complex. The recognition of palindrome structure may reduce the possibility of bacteriophage escape due to sequence mutations.**

Reviewer #3 (Remarks to the Author):

This reviewer is mostly satisfied by the substantial new experimental data and analysis provided by the authors in this revised version of the manuscript. Additional evidence for the folding and enzymatic activity of Gabija is now included that strengthens the original draft.

Response: We thank the reviewer's the constructive comments for strengthening our manuscript.

The plaque assay presented in Extended Data Figure 2 appears possibly overloaded and 7-8 logs of protection is almost unbelievable. Trusting that all sample lanes were loaded the same, the phenotype is remarkable.

Response: As per your suggestion, we have conducted new sets of T7 phage plaque assays and control the growth time carefully to prevent over growth. In addition, we have also conducted T2 phage plaque assays and included this result in the source data.

A minor concern is that there is limited utility of the color scheme used in Extended Data Figure 8. Consider another version that may rely less on readers ability to distinguish between closely related or otherwise complementary colors.

Response: As per your suggestion, we have redrawn Extended Data Figure 8 using comparable colors scheme.